# The Role of Macrophage in the Pathogenesis of Osteoporosis

**DOI:** 10.3390/ijms20092093

**Published:** 2019-04-28

**Authors:** Deng-Ho Yang, Meng-Yin Yang

**Affiliations:** 1Division of Rheumatology/Immunology/Allergy, Department of Internal Medicine, Taichung Armed-Forces General Hospital, Taichung 411, Taiwan; 2Department of Medical Laboratory Science and Biotechnology, Central Taiwan University of Science and Technology, Taichung 406, Taiwan; 3Division of Rheumatology/Immunology/Allergy, Department of Internal Medicine, Tri-Service General Hospital, National Defense Medical Center, Taipei 114, Taiwan; 4Department of Neurosurgery, Jan-Ai General Hospital, Taichung 412, Taiwan; 5Department of Neurosurgery, Tri-Service General Hospital, National Defense Medical Center, Taipei 114, Taiwan; 6College of Nursing, Central Taiwan University of Science and Technology, Taichung 406, Taiwan; 7Department of Neurosurgery, Taichung Veterans General Hospital, Taichung 407, Taiwan

**Keywords:** osteoporosis, macrophage, cytokine, chemokine, estrogen

## Abstract

Osteoporosis is a systemic disease with progressive bone loss. The bone loss is associated with an imbalance between bone resorption via osteoclasts and bone formation via osteoblasts. Other cells including T cells, B cells, macrophages, and osteocytes are also involved in the pathogenesis of osteoporosis. Different cytokines from activated macrophages can regulate or stimulate the development of osteoclastogenesis-associated bone loss. The fusion of macrophages can form multinucleated osteoclasts and, thus, cause bone resorption via the expression of IL-4 and IL-13. Different cytokines, endocrines, and chemokines are also expressed that may affect the presentation of macrophages in osteoporosis. Macrophages have an effect on bone formation during fracture-associated bone repair. However, activated macrophages may secrete proinflammatory cytokines that induce bone loss by osteoclastogenesis, and are associated with the activation of bone resorption. Targeting activated macrophages at an appropriate stage may help inhibit or slow the progression of bone loss in patients with osteoporosis.

## 1. Introduction

Osteoporosis is a systemic skeletal disorder characterized by a generalized increase in bone fragility that results in fractures of the hip, spine, or wrist. There are many risk factors associated with the progression of osteoporosis, including advanced age, low body-mass index, long-term glucocorticoid therapy, history of smoking, family history of hip fracture, excessive alcohol intake and previous fragility fracture. Osteoporosis occurs in all populations with a greater prevalence in postmenopausal women [1]. Women with osteoporosis have poorer musculoskeletal status than women without osteoporosis [2]. Osteoporosis is associated with limitations of daily activities, an increase in the occurrence of falls, and a consequent increased risk of fracture. The etiology of osteoporotic fractures is complex and the incidence of osteoporotic fractures is high among people aged 50 to 54 years [3]. A higher mortality is observed in these patients after osteoporotic fractures occur [4]. Osteoporotic fracture is associated with significant morbidity, mortality, poor quality of life, and increasing health care costs. Therefore, the pathogenesis of bone loss or fracture is an important issue for the prevention of osteoporosis. 

## 2. Osteoblast, Osteoclast and Osteocyte in Bone Formation and Homeostasis

Osteoblasts originate from the mesenchymal stem cells and are important for the progression of bone formation. Co-expression of bone-specific alkaline phosphatase, type I collagen, and non-collagenous matrix proteins is observed in mature osteoblasts. Transforming growth factor-β (TGF-β) has an important role in bone formation by enhancing osteoblast proliferation [5]. TGF-β can also block osteoblast apoptosis and recruit osteoblastic precursors to the bone surface [6,7,8]. interleukin (IL)-4, IL-10, IL-13, and IL-18 may induce elevation of osteoprotegerin (OPG) and reduction of receptor activator of NF-κB ligand (RANKL) [9]. Final differentiation of osteoblasts can form osteocytes, which become embedded in the mineralized matrix. The processes of bone formation by osteoblasts are enhanced by administration of vitamin D and parathyroid hormone. The Wnt/β-catenin is important for osteoblastic differentiation in skeletal biology and disease [10]. Dickkopf-1 and sclerostin are Wnt inhibitors and can regulate the expression of Wnt/β-catenin in osteoblasts [11]. More than 90% of bone cells are osteocytes, which release chemicals to the bone surface that attract osteoclasts. Osteocytes may secrete sclerostin to limit further bone formation by osteoblasts, and play a major role in matrix mineralization [12]. Bone remodeling involves coupling and regulation of osteoclasts and osteoblasts. Besides these resorptive and formative cells, T-cells, B-cells and macrophages may also influence the immune system and bone loss [13]. The bone remodeling cycle is shown in Figure 1.

## 3. Macrophages in the Pathogenesis of Osteoporosis

In the process of bone remodeling, different states of resorption, reversal, and formation are found. Activated monocytes or bone marrow macrophage precursors adhere to the bone surface to form multinucleated osteoclasts [14]. The bone remodeling cycle is regulated by local and systemic factors. Osteoclasts and osteoblasts are both important for the pathogenesis and progression of osteoporosis. Osteoclasts induce bone resorption and osteoblasts are associated with bone formation. Normal bone quality involves a neutral balance between resorption and formation. A multinucleated osteoclast is differentiated from the mononuclear osteoclast precursor of hemopoietic stem cell. RANKL and macrophage-colony stimulating factor (M-CSF) can induce the proliferation and activation of osteoclasts via the receptor, RANK [15,16]. Therefore, the balance between resorption and formation determines the progression of osteoporosis. M1 macrophages are associated with exacerbation of inflammation and express proinflammatory cytokines. M2 macrophages are associated with anti-inflammatory reactions through the expression of anti-inflammatory cytokines [17]. When macrophages are exposed the stimulation of RNAKL, macrophages may induce osteolcastogenesis and lead to increased M1/M2 ratio in ovariectomized mice. Estrogen can protect M2 macrophage from RANKL stimulation through estrogen receptor αand the downstream blockage of NF-κB p65 nuclear translocation [18]. The blocking of estrogen deficiency-mediated M2 macrophage osteoclastogenesis by reducing the M1/M2 ratio may be a potential therapeutic target in treating postmenopausal osteoporosis. Macrophages play a major role in the activation and formation of osteoclasts and are differentiated from monocytes via M-CSF [19]. The different presentations of macrophages in different organs are shown in Figure 2. Besides the differentiation of osteoclasts from the macrophage lineage, macrophage precursors also differentiate into monocytes, macrophages, and dendritic cells. The activation of macrophages may induce the elevation of interferon-γ (IFN-γ), IL-1, tumor necrosis factor-α (TNF-α), complement proteins, and prostaglandins levels [20]. Macrophages are important for the pathogenesis of osteoporosis [19]. 

## 4. The Cytokines from Macrophages Contribute to the Process of Osteoporosis

Macrophages play a major role in the innate and adaptive immune system. Differentiation of macrophages can be found in various tissues including liver, lung, brain, and bone marrow. Macrophages are divided into M1, which express proinflammatory mediators, and M2 that are involved in anti-inflammatory reactions [21]. In the innate immune system, macrophages can execute phagocytosis and opsonization [22]. Different cell receptors of CD14, Fcγ, and CD25 are found in macrophages. Major histocompatibility complex class II molecules and CD23 are also found. These receptors are important for the progression of phagocytosis. IL-4 and IFN-γ can regulate the different functions of macrophages. Macrophages, in turn, may be activated to induce IL-6, TNF-α, IFN-γ, complement protein, and prostaglandins in the immune system. The cytokines expressed by macrophages associated with stimulation or inhibition of osteoclastogenesis include IL-6, IL-18, IL-23, IL-27, and TNF-α [23]. IL-6, a proinflammatory cytokine, can activate osteoclastogenesis [24]. During the inflammation state, proinflammatory cytokines including TNFα, IL-1β, and IL-6, may promote the differentiation and activation of osteoclasts [25]. IL-18 secreted via macrophages may regulate the Th1 differentiation and the IFN-γ production, and is an inhibitor of the TNF-α mediated osteoclastogenesis [26]. Among osteoporotic women, decreased levels of serum IL-18 binding protein and elevated levels of serum IL-18 are observed [27,28]. IFN-γ has a dual role in osteoclasts including the promotion of osteoblast differentiation and inhibition of bone marrow adipocyte formation in different stages [29]. IFN-γ can activate macrophages, but macrophages can secrete IL-18 to regulate the IFN-γ production. IL-23 has been shown to activate osteoclasts [30]. Adding IL-23 to bone marrow stromal cells led to an increased differentiation towards the osteoblast lineage [31]. In the femur of a rat osteoporosis model, IL-23 is reduced after adequate estrogen therapy for improvement of bone mineral density [32]. IL-27 may suppress the expression of RANKL in Th17 cells and CD4+ T cells [33]. IL-27 also inhibited osteoblast apoptosis through increased Egr-2 expression [34]. TNF-α from macrophages may induce indirect osteoclastic activation through RANKL in bone remodeling [35]. Therefore, inflammatory arthritis, such as rheumatoid arthritis, can induce progressive bone loss when the disease is poorly controlled. The different cytokines of macrophages associated with osteoporosis are shown in Figure 1. 

## 5. The Fusion of Macrophages/Monocytes to Form Multinucleated Cells—Osteoclasts

Macrophages have the ability to fuse and develop into multinucleated cells during an acute infection and inflammation state. The formation of granuloma induced by tuberculosis infection and vasculitis may result in these multinucleated giant cells. In systemic inflammation, macrophages can release reactive oxygen and reactive nitrogen species to induce the formation of multinucleated giant cells. M-CSF is the most important cytokine in the initial stage of macrophages differentiation from hematopoietic stem cells. Different phases are observed during the formation of multinucleated cells. RANKL is the major cytokine responsible for the stimulation of osteoclasts into mature multinucleated osteoclasts [36]. The cytokines, IL-4 and IL-13, may induce macrophages to form multinucleated giant cells during the course of bone resorption [37,38,39]. The proliferation and differentiation of macrophages may be stimulated by M-CSF. After the stimulation via M-CSF, RANKL may activate the proliferation of osteoclasts. Fusion-competent osteoclasts may be induced by RANKL. Multinucleated giant cells originate from this fusion of cells, which develops to form multinucleated osteoclasts or giant cells [19]. The fusion of macrophages to form multinucleated osteoclasts is shown in Figure 1. 

## 6. Different Cytokines for the Pathogenesis of Osteoporosis

RANKL can be expressed by different cells including T cells, B cells, bone-marrow stromal cells, and bone-forming osteoblasts. Mice with depletion of *RANKL* gene show severe osteopetrosis and lack mature circulating osteoclasts [40]. The differentiation of osteoclasts may be inhibited by the decoy receptor OPG, which is produced by osteoblasts [41]. Proinflammatory cytokines including IL-1 and TNF-α can stimulate osteoclastogenesis in vitro [42]. Other osteoclastogenic cytokines include IL-6, IL-8, IL-15, IL-17, and IFN-γ [9,43]. High dosage of IFN-γ may promote the differentiation of osteoclasts, and the effect of bone loss is enhanced in situations of estrogen deficiency [44,45]. The immune response in osteoclastogenesis via IFN-γ include activation of RANKL/RANK pathway and promotion of fused mononucleated osteoclasts [29]. In patients with rheumatoid arthritis (RA), activated T cells can directly trigger osteoclastogenesis through RANKL/RANK/OPG pathway [46,47]. Therefore, juxta-articular osteopenia of both hands and osteoporotic fracture are usually found during the disease course of RA. The role of T cells in regulating osteoclastogenesis is associated with the formation of osteoclasts. B cells may participate in osteoclastogenesis by expression of RANKL for osteoclast differentiation and serve as osteoclast progenitors [48]. Osteoclast-associated receptor may be expressed by macrophages or monocytes in order to modulate the innate and adaptive immune response [49]. 

## 7. Estrogen Deficiency Induced the Expression of Different Cytokines in Osteoporosis

Estrogen can directly inhibit osteoclastic bone resorption by inducing apoptosis of osteoclasts [50]. Estrogen may induce osteoblast differentiation in bone formation by binding the estrogen receptor through the upregulation of PACE4 expression [51], and it also has an anabolic effect on the function of osteoblasts [52]. Estrogen serves different biological functions in the regulation of osteogenic differentiation with involvement of the Wnt/β-catenin signaling pathway [53]. Estrogen loss may also influence the immune system through upregulation of T and B cells [54]. Higher expression of circulating IL-1, IL-7, and IFN-γ are found in patients with estrogen withdrawal [55,56]. Estrogen deficiency can stimulate T-cell activation and production of pro-osteoclastogenic cytokines. The levels of follicle-stimulating hormone (FSH) are increased during the development of estrogen deficiency. FSH receptors are present on osteoclasts, osteoclast precursors, and mesenchymal stem cells, and promote osteoclast differentiation, activity, and survival [57]. The net effect of estrogen deficiency on the bone is an increased activation of bone remodeling and osteoclasts. The bone loss induced by estrogen deficiency has a complex mechanism with predominant involvement of the immune system rather than a direct action of estrogen on bone cells [56]. The possible mechanism underlying the association of estrogen and bone loss is shown in Figure 3. Therefore, estrogen deficiency is associated with bone loss by influencing activity and formation of osteoclasts or proliferation of osteoblasts.

## 8. The Activation and Differentiation of Macrophages to Osteoclasts in the Development of Osteoporosis

The differentiations of osteoclasts are both from hematopoietic precursor cells and macrophage lineage [58]. Osteoclastogenesis from macrophages is activated by M-CSF and RANKL, and the blockage of RANKL signaling pathway may prevent the progression of osteoporosis in mice models [59,60]. The bone loss in ovariectomized mice is also associated with osteoclast differentiation of bone marrow-derived macrophages [61]. The expression of TNF receptor associated factor (TRAF) 6 and TRAF3 are both important in the differentiation of early osteoclasts in osteoclast’s precursors and macrophages. The level of TRAF3 protein decreases in bone and bone marrow with aging [62]. TRAF3 has been revealed to be a powerful negative regulator in B cells [63]. Proliferation of B cells can induce the expression of RANKL. Therefore, TRAF3 may be a target for the prevention of immune related bone loss.

M1 macrophages can induce exacerbation of inflammation and are associated with the development of osteoporosis. Bisphosphonates are used for the treatment of osteoporosis, and associated osteonecrosis of the jaw is an unusual complication. The related osteonecrosis may be due to an abnormal activation of M1 macrophages [64]. The differentiation of osteoclast from macrophage can be activated by M-CSF and RANKL, and RANKL signaling pathway activates the major regulator of NFATc1 [65]. Down-regulating NFATc1 may inhibit RANKL-mediated osteoclastogenesis [66]. 

## 9. The Effect of Macrophages in Osteoblasts

Osteoblasts are the major cells responsible for bone formation and originated from mesenchymal stem cells. The osteoblastogenesis of mature osteoblasts is controlled by different transcription factors. Mature osteoblasts may differentiate to osteocytes and lining cells. In the cycle of bone remodeling, macrophages have a major role in the induction of osteoclastogenic differentiation during the state of resorption. The differentiation of osteoclasts serves as a regulatory step in the formation of osteoblasts and osteocytes (Figure 1). However, young macrophage cells present during the rejuvenation process and are associated with bone repair in mice [67]. Adequate activation of osteoblasts may be found during the progression of bone repair. Activated macrophages are the most likely candidates to promote bone formation and have also been implicated in tissue repair processes [17]. Osteoblasts are important for the progression of bone formation under adequate activation and proliferation. IL-4 and IL-13 can recruit and activate osteoblasts for the progression of bone resorption and healing [68,69]. IL-4 and IL-13 may inhibit osteoclast differentiation and bone resorption via activation of the receptors on osteoblasts and osteoclasts that affect the RANKL/RANK/OPG system [70]. In osteoblasts, IL-4 and IL-13 can suppress the production of prostaglandin through the induction of IL-1. TNF-α has the effect of stimulating osteoblast chemotaxis in vitro [71]. However, osteoblasts are regulated by different cytokines during the phase of bone resorption. IL-6 may inhibit osteoblast differentiation with disrupting the balance of healthy bone turnover [72]. TNF-α may suppress osteoblast differentiation in RA patients [73]. Inflammatory cytokines can activate osteoclasts and promote bone resorption. At the same time, inhibition or attenuation of the osteoblasts is also observed. Therefore, adequate activation and regulation of osteoblasts are important for the pathogenic presentation of bone remodeling. The balance between bone formation and resorption is regulated by different cytokines. 

## 10. The Activation and Regulation of Chemokines in Macrophages-Associated Osteoporosis

Chemokines are important for the migration of circulating hematopoietic cells into different tissues. CXCL-8 and CCL-20 are elevated in inflammatory arthritis, and they may enhance osteoblast-mediated osteoclastogenesis through the production of IL-6 [74]. The active expression of CCL-6 is observed during the differentiation of osteoclasts, and can progress to bone loss in vitro [75]. CCL-4 can inhibit migration of osteoclast precursor cells from the bone marrow into the bone surfaces, and cannot affect the differentiation of osteoclasts [76]. Elevation of CXCL-10 is found during osteoclast differentiation, and CXCL-10-neutralizing antibodies can reduce the effect of osteoclastogenesis [77,78]. Higher expression of CX3CR-1 is found on osteoclast precursor cells, and CX3CL-1 from osteoblasts can bind the receptor of CX3CR-1 and induce migration and adhesion of osteoclasts to the bone marrow [79]. During activation of osteoclasts, expression and activation of CCL-3 is observed in vitro [75]. Therefore, the chemokines associated with bone resorption include CXCL-8, CCL-20, CCL-6, CCXL-10, CCL-3, while CCCL-4 is associated with bone formation. Among the inflammatory progression, most chemokines serve as a mediator for osteoclastogenics. 

## 11. The Effect of Macrophages in Osteocytes

The role of osteocytes is important in the bone remodeling cycle and may be regulated by osteoclasts and osteoblasts. The differentiation of osteocytes is influenced by osteoblasts. Osteocytes form the major structures of cortical and calcaneus bone and have different physiological function for bone resorption or formation. Osteocytes promote the production of RANKL and decrease OPG expression. Consequently, the RANKL/OPG ratio increases, osteoclastogenesis occurs, and the enhancement of bone resorption in the unloading activity is seen. The loss of RANKL in osteocytes may increase cancellous bone mass in the osteogenesis imperfecta mouse model [80]. Activation of macrophages can induce the production of proinflammatory cytokines including IL-1, IL-6 and TNF-α, which may be important for the inflammatory bone loss [81]. After stimulation of IL-1 and TNF-α, active fibroblast growth factor-23 secretion by osteocytes may contribute to hypophosphataemia during sepsis [82]. IL-1 may promote loss of osteocyte viability through NF-kB/RANKL signaling [83]. Soluble IL-6 may promote bone formation or osteoclastogenesis in different levels among the progression of normal bone growth and remodeling [84]. The osteoclastogenesis-supporting activity is reduced in zoledronate-treated osteocyte-like MLO-Y4 cells in the presence of IL-6 neutralizing therapy [85]. Bisphosphonate-related osteonecrosis of the jaw may be progressing by this pathway. Immunohistochemical staining of TNF-α, IL-6, and sclerostin in osteocytes are statistically higher in the patient with spinal cord injury [86]. Macrophages may inhibit osteocyte viability through the effect of TNF-α and IL-6 in the model of monosodium urate crystal-induced inflammation [87]. During the stage of bone repair, expression of IL-18 from macrophages may serve an important role by increasing expression during bone formation or decreasing expression in the model of bisphosphonate-related osteonecrosis of the jaw [88,89]. Osteocytes may be regulated by macrophage secreting inflammatory cytokines (TNF-α, IL-6) to influence bone turnover. However, osteocytes suppress sclerostin expression with the consequent induction of Wnt signaling and enhanced bone formation in the loading activity. The physiological effect of osteocyte-intrinsic mTORC1 signaling may decrease trabecular bone mass [90]. In different conditions, osteoclastogenesis or osteoblastogenesis influence bone loss. In animal models, N-methyl pyrrolidone may prevent estrogen deficiency, induce increased expression of sclerostin, and decrease the progression of osteoporosis [91]. The stimulation of bone formation is induced after anti-sclerostin antibody treatment in ovariectomized rats [92,93]. Sclerostin from osteocytes may suppress bone formation by decreasing osteoblast-related autophagy or apoptosis in multiple myeloma-related osteolytic bone diseases [94]. In the mouse model of estrogen deficiency, the condition of autophagy in osteocytes is associated with oxidative stress [95]. Aging, estrogen deficiency, and steroid may stimulate oxidative stress and induce the abnormal apoptosis of osteocytes and osteoblasts with consequent progression of osteoporosis or bone loss [96]. IFN-β is an anti-osteoclastogenic cytokine that activates the signaling of Toll-like receptor 5 and can be secreted by osteocytes. Adequate activation and proliferation of osteocytes are important for bone formation. Blocking and inhibiting the abnormal apoptosis of osteocytes by low-dose risedronate can prevent bone loss in ovariectomized rats [97]. 

## 12. The Modulation of Insulin-Like Growth Factor-1 (IGF-1) in Macrophages

During inflammation, macrophages are the major producers of cytokines. Activated macrophages are able to promote osteoclastogenesis for the progression of osteoporosis. IGF-1 can inhibit the activation of macrophages under the stimulation of IL-4 [98]. Overexpression of IGF-1 may delay the secretion of proinflammatory cytokines by macrophages in the model of skeletal muscle injury [99]. The limitation of activated macrophages by IGF-1 is through the PI3K/Akt signaling pathway [100]. IGF-1 serves as a key factor for anti-inflammation processes and macrophage modulation in models of murine skeletal muscle [101]. IGF-1 may activate the differentiation of osteoblasts and promote bone formation [102]. Therefore, IGF-1 has different effects during bone remodeling. IGF-I may stimulate both markers of bone formation and bone resorption, but a low dose of IGF-1 can promote bone formation [103]. Elevation of parathyroid hormone (PTH) is associated with the activation of bone turnover and is important to regulate bone remodeling. PTH receptor signaling can increase the RANKL/OPG ratio in osteoblasts and osteocytes. Therefore, primary and secondary hyperparathyroidism may increase the risk of osteoporosis related fracture. T lymphocytes express the PTH receptor and are essential for the development of osteoclastogenesis [104]. The direct effects of PTH on osteoblasts and osteocytes, and indirect actions on osteoclasts, promote both bone formation and bone resorption [105]. The interaction between T cells and macrophages is associated with osteoimmunology. The administration of PTH has different dose-dependent effects including bone formation or osteoclastic callus remodeling during fracture healing [106]. 

## 13. The Possible Therapy of Osteoporosis by Targeting Macrophages

Osteoporosis is treated by different traditional drugs. Calcium and vitamin D supplementation are used in the prevention and treatment of osteoporosis. Hormone replacement therapy is used in postmenopausal women with osteoporosis. Estrogen agonist-antagonists have antiresorptive effects on the bone and can reduce the risk of vertebral fractures in postmenopausal women. Bisphosphonates are most commonly used for the therapy of osteoporosis due to their effects in the suppression of osteoclast-mediated bone resorption. Strontium can compete with calcium to stimulate osteoblastic activity and decrease osteoclastic activity in the course of bone turnover. Parathyroid hormone therapy can increase bone formation and mass through parathyroid hormone-related protein receptor signaling. Denosumab is a fully human monoclonal antibody that binds RANK and inhibits the activity of RANKL. Decreasing the levels of RANKL may decrease the activity of osteoclasts. 

The action of TRAF3 may prevent osteoporosis-related bone loss in inflammatory osteoporosis. RANKL can enhance TNF-induced osteoclast formation independently of TRAF6 and by degrading TRAF3 [107]. TRAF3 may be an agent to prevent bone loss. Increased expression of sclerostin is found during bone remodeling along with the elevation of RANKL, and inhibition of sclerostin expression may improve bone loss. Anti-sclerostin may serve as an agent for bone formation. Macrophages are activated to induce the progression of bone resorption during the cycle of bone remodeling. Adequate blockage of macrophage activation may suppress the progression of bone resorption and the pathogenesis of osteoporosis. Macrophages proliferate via M-CSF, and the adequate anti-M-CSF treatment may suppress the proliferation of macrophages. However, macrophages also have an important role in regulating the quality of bone repair. In patients with RA, tocilizumab adequately inhibits circulating IL-6, which leads to improved bone loss in osteoporosis [108]. Abatacept that interferes with T cell functioning, may increase bone mineral density at the femoral neck in patients with RA [109]. When the systemic inflammation is controlled by different biologics (including anti-TNF-α, anti-IL-6, anti-IL-1, T and B cell regulators), these anti-cytokine therapies are associated with the improvement of bone loss [110]. Macrophages were stimulated to promote osteoclastogenesis via M-CSF, TNF-α, IL-1β, IL-6, IL-17, and IL-23 [111]. When macrophage-related cytokines, which include IL-1β, IL-6, and TNF-α, are regulated, the progression of bone loss is improved. When the functions of B and T cells are regulated, the suppression of macrophage may develop through the inhibition of IFN-γ. Therefore, osteoporosis-related bone loss is prevented or limited under the suppression of macrophages or inhibition of macrophage-related cytokines. IL-23 expression by macrophages can induce osteoclastogenesis with bone destruction during inflammation. Monotherapy against IL-23 has been used to manage post-menopausal bone loss [31]. IL-27 from macrophages can suppress the expression of RANKL and inhibit the apoptosis of osteoblasts. Treatment with IL-27 agent prevented the loss of trabecular micro-architecture and preserved cortical bone parameters [34]. However, macrophages have other effects on bone formation during fracture-related bone repair. Adequate activated macrophages are important for new bone formation. 

## 14. Conclusions

Activated macrophages may secrete proinflammatory cytokine that induces bone loss by osteoclastogenesis, and are associated with activation of bone resorption. Furthermore, adequate activation of macrophages is needed during the state of bone repair and is also associated with bone formation. Different expression of activated macrophages including proinflammatory mediators in M1 and anti-inflammatory reactions in M2, can regulate the progression of bone resorption and bone formation. Osteoporosis is always associated with systemic inflammation via the abnormal activation of osteoclasts. Adequate inhibition of macrophages may improve the systemic inflammation associated with osteoporosis. Macrophages can affect osteoclasts, osteoblasts, and osteocytes during the progression of bone loss. Regulation of proinflammatory cytokine from macrophages may inhibit osteoclastogenesis. Adequate stimulation of IL-27 from macrophages may inhibit osteocalstogenesis. IL-4 and IL-23 may induce macrophages to form multinucleated giant cells during the phase of bone resorption. The pharmaceutical targets in macrophage polarization or macrophage related cytokine dysregulation may be an important issue to develop in the future. 

## Figures and Tables

**Figure 1 ijms-20-02093-f001:**
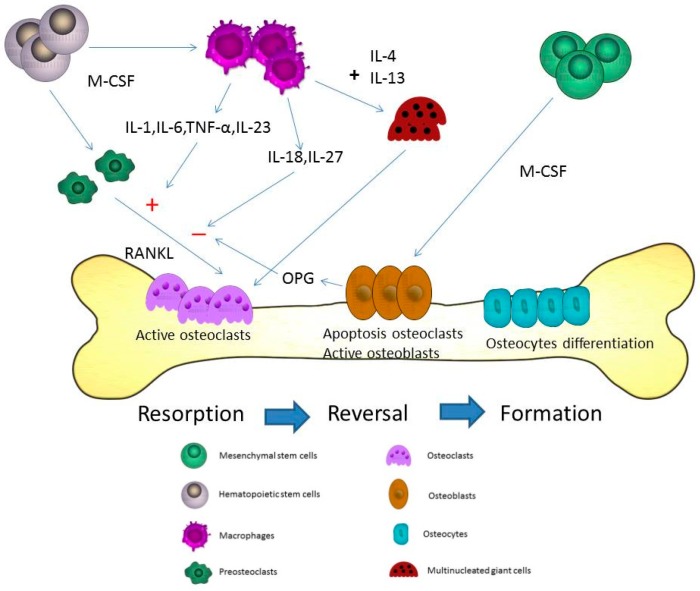
Microdamages in the bone-remodeling units of cancellous or cortical bone induced by osteoclasts. Osteoclasts may be activated by different cytokines including receptor activator of NF-κB ligand (RANKL), interleukin (IL)-1, IL-6, and macrophage-colony stimulating factor (M-CSF) in the resorption state. After the resorption process, the reversal state progresses. In the reversal state, apoptosis of osteoclasts may be induced to stop the bone resorption. The replacement of osteoblasts is observed at the same time. The activated osteoblasts refill the resorption pits and tunnels on the bone surface. In the formation state, osteoblasts directly adhere to the bone surface and progressively form into osteocytes. The proliferation of osteocytes can use these lacuna-canalicular networks to connect within the bone matrix. The cytokines from macrophages including IL-6, tumor necrosis factor-α (TNF-α), IL-23, IL-18, and IL-27 can induce and inhibit osteoclastogenesis through RANKL in bone remodeling. M-CSF is the most important cytokine in the initial stage of macrophage differentiation from hematopoietic stem cells. Different phases are observed during the formation of the multinucleation. RANKL is the major cytokine that stimulates osteoclasts into mature multinucleated osteoclasts. The cytokines, IL-4 and IL-13, may induce macrophages to form multinucleated giant cells during the course of bone resorption.

**Figure 2 ijms-20-02093-f002:**
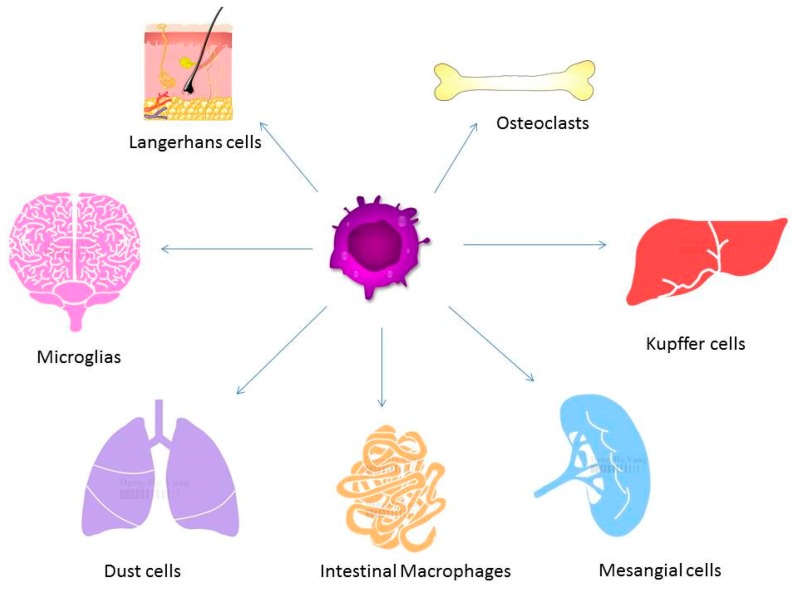
Macrophages exist in different tissues including lung, liver, and brain and have different functions. Different forms of macrophages include Kupffer cells in the liver, alveolar macrophages in the lung, osteoclasts in the bone, and microglia in the brain.

**Figure 3 ijms-20-02093-f003:**
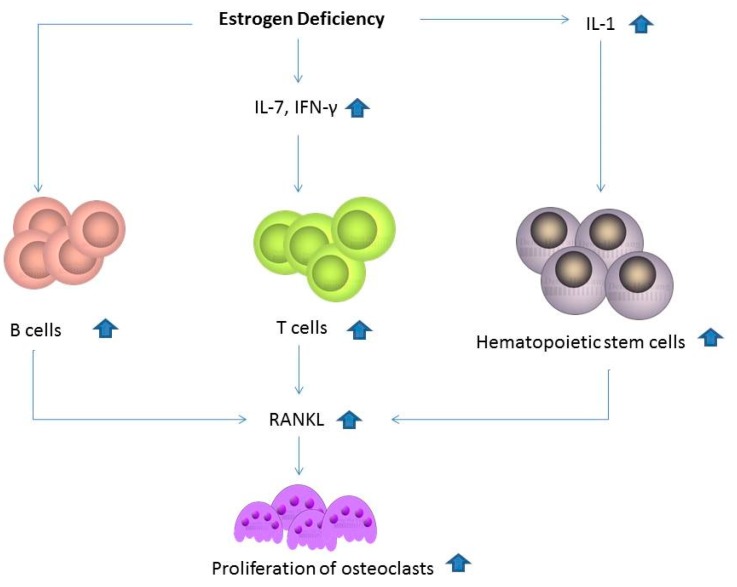
Estrogen loss may also influence the immune system by the upregulation of T and B cells. Higher expression of circulating IL-1, IL-7, and IFN-γ is found in patients with estrogen withdrawal. Estrogen deficiency can stimulate T-cell activation and production of pro-osteoclastogenic cytokines.

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
