# Peer review of "The Role of Macrophage in the Pathogenesis of Osteoporosis"

_ijms, 2019, doi:10.3390/ijms20092093_

Round 1
Reviewer 1 Report
This is a review regarding the role of macrophages in the pathogenesis of osteoporosis. There are already several good reviews on this topic published in PubMed listed journals, so a special novel focus on the topic or novel findings need to be included in order to make the review original.
The authors try to include many aspects on how macrophages are involved in the pathogenesis of osteoporosis, which unfortunately often results in a confusing listing of factors secreted by different cell types without proper explaining why they are important for osteoporosis and what are the underlying molecular mechanisms explaining their effects. The review should focus on macrophages in osteoporosis –as the title implies- and not include role and function of immune cells as T- and B-cells. Also, the manuscript falls short to closer elaborate on effects and role of macrophage subsets (MO, M1, M2) and their polarization status on bone turnover and regeneration after trauma under a compromised condition as osteoporosis.
All in all, the manuscript needs to be rewritten with clearer focus, meaningful organization of topics and paragraphs, and deeper insight into mechanisms of how macrophages are involved into osteoporosis pathology.
The manuscript needs extensive language revision as often wrong wording or strange terms or meaningless headlines are used (some examples see below)!
Specific points
- Page 5, lanes 148-163: Here, function of osteoblasts and osteocytes are described. This does not belong into the context of his paragraph, which is supposed to describe the contribution of cytokines to osteoporosis. It should be placed into an extra paragraph shortly describing the cell types, which are relevant for bone formation and homeostasis.
- The figures should be numbered in the order of appearance in the text, e.g. figure 1 before figure 2.
- Page 2: Paragraph “Expression of macrophages in the pathogenesis of osteoporosis” First, meaning of “expression” in this context is not correct. Second, references are missing for this paragraph.
- Page 7: Paragraph headline “The cytokines between macrophage and …..” Again, wrong wording. Lane 214: “In macrophages, adequate activation of osteoblasts….” Wrong wording!
- Page 7, lane 241: osteoclasts do not proliferate anymore. Please correct.
- Page 7-8: paragraph: “The effect of cytokines in osteocytes” Where is the connection to macrophages?
- Page 8: Headline has no meaning! What is meant with “….targeting macrophage associated medicine”?
- Page 9, lane 319: What mans “… and TNF-a are regulated and the progression of bone loss is improved?
- Page 9, lane 320: What means “When T- and B- cells are regulated, the activated macrophages are suppressed”??
- Page 9, lanes 324-326: If Il-27 inhibits apoptosis of osteoblasts how can anti-IL-27 prevent loss of bone and preserve bone?
- Figure 1: osteocytes do not proliferate anymore they are terminally differentiated cells!
Author Response
This is a review regarding the role of macrophages in the pathogenesis of osteoporosis. There are already several good reviews on this topic published in PubMed listed journals, so a special novel focus on the topic or novel findings need to be included in order to make the review original.
The authors try to include many aspects on how macrophages are involved in the pathogenesis of osteoporosis, which unfortunately often results in a confusing listing of factors secreted by different cell types without proper explaining why they are important for osteoporosis and what are the underlying molecular mechanisms explaining their effects. The review should focus on macrophages in osteoporosis –as the title implies- and not include role and function of immune cells as T- and B-cells. Also, the manuscript falls short to closer elaborate on effects and role of macrophage subsets (MO, M1, M2) and their polarization status on bone turnover and regeneration after trauma under a compromised condition as osteoporosis.
Response:
We had corrected and focus the discussion about the different role of macrophage in the pathogenesis of osteoporosis and showed as below: In the process of bone remodeling, different states of resorption, reversal, and formation are found. Activated monocytes or bone marrow macrophage precursors adhere to the bone surface to form multinucleated osteoclasts [14]. The bone remodeling cycle is regulated by local and systemic factors. Osteoclasts and osteoblasts are both important for the pathogenesis and progression of osteoporosis. Osteoclasts induce bone resorption and osteoblasts are associated with bone formation. Normal bone quality involves a neutral balance between resorption and formation. A multinucleated osteoclast is differentiated from the mononuclear osteoclast precursor of hemopoietic stem cell. Receptor activator of NF-κB ligand (RANKL) and macrophage-colony stimulating factor (M-CSF) can induce the proliferation and activation of osteoclasts via the receptor, RANK [15, 16]. Therefore, the balance between resorption and formation determines the progression of osteoporosis. M1 macrophages are associated with exacerbation of inflammation and express proinflammatory cytokines. M2 macrophages are associated with anti-inflammatory reactions through the expression of anti-inflammatory cytokines [17]. When macrophages are exposed the stimulation of RNAKL, macrophages may induce osteolcastogenesis and lead to increased M1/M2 ratio in OVX mice. Estrogen can protect M2 macrophage from RANKL stimulation through estrogen receptor αand the downstream blockage of NF-κB p65 nuclear translocation [18]. The blocking of estrogen deficiency-mediated M2 macrophage osteoclastogenesis by reducing the M1/M2 ratio may be a potential therapeutic target in treating postmenopausal osteoporosis. Macrophages play a major role in the activation and formation of osteoclasts and are differentiated from monocytes via M-CSF [19]. The different presentations of macrophages in different organs are shown in Figure 2. Besides the differentiation of osteoclasts from the macrophage lineage, macrophage precursors also differentiate into monocytes, macrophages, and dendritic cells. The activation of macrophages may induce the elevation of interferon-γ (IFN-γ), interleukin (IL)-1, tumor necrosis factor-α (TNF-α), complement proteins, and prostaglandins levels [20]. Macrophages are important for the pathogenesis of osteoporosis [19]. See page 2, line 66-90
Page 5, lanes 148-163: Here, function of osteoblasts and osteocytes are described. This does not belong into the context of his paragraph, which is supposed to describe the contribution of cytokines to osteoporosis. It should be placed into an extra paragraph shortly describing the cell types, which are relevant for bone formation and homeostasis.
Response:
We had added other paragraph as below: Osteoblast, osteoclast and osteocyte in bone formation and homeostasis. See page 2, line 47-63
The figures should be numbered in the order of appearance in the text, e.g. figure 1 before figure 2.
Response:
We had corrected this inappropriate appearance.
Page 2: Paragraph “Expression of macrophages in the pathogenesis of osteoporosis” First, meaning of “expression” in this context is not correct. Second, references are missing for this paragraph.
Response:
We had corrected the title as below: Macrophages in the pathogenesis of osteoporosis, and added references in this paragraph. See page 2, line 65
Page 7: Paragraph headline “The cytokines between macrophage and …..” Again, wrong wording. Lane 214: “In macrophages, adequate activation of osteoblasts….” Wrong wording!
Response:
We had corrected the wrong word as below: The effect of macrophages in osteoblasts. See line 217. Adequate activation of osteoblasts may be found during the progression of bone repair. See page 7, line 224-225
Page 7, lane 241: osteoclasts do not proliferate anymore. Please correct.
Response:
We had corrected as below: During activation of osteoclasts, expression and activation of CCL-3 is observed in vitro. See page 8, line 251-252.
Page 7-8: paragraph: “The effect of cytokines in osteocytes” Where is the connection to macrophages?
Response:
We had corrected the paragraph as below: The effect of macrophages in osteocytes.
Activation of macrophages can induce the production of proinflammatory cytokines including IL-1, IL-6 and TNF-α, which may be important for the inflammatory bone loss [79]. After stimulation of IL-1 and TNF-α, active fibroblast growth factor-23 secretion by osteocyte may contribute to hypophosphataemia during sepsis [80]. IL-1 may promote loss of osteocyte viability through the NF-kB/RANKL signaling [81]. Soluble IL-6 may promote bone formation or osteoclastogenesis in different levels among the progression of normal bone growth and remodeling [82]. The osteoclastogenesis-supporting activity is reduced in zoledronate-treated osteocyte-like MLO-Y4 cells in the presence of IL-6 neutralizing therapy [83]. Bisphosphonate-related osteonecrosis of the jaw may be progressing by this pathway. Immunohistochemical staining of TNF-α, IL-6, and sclerostin in osteocytes are statistically higher in the patient with spinal cord injury [84]. Macrophages may inhibit osteocyte viability through the effect of TNF-α and IL-6 in the model of MSU crystal-induced inflammation [85]. During the stage of bone repair, expression of IL-18 from macrophage may severe as important role by increasing expression during bone formation or decreasing expression in the model of bisphosphonate-related osteonecrosis of the jaw [86, 87]. Osteocytes may be regulated by macrophage secreting inflammatory cytokines (TNF-α, IL-6) to influence bone turnover. See page 8, line 263-278.
Page 8: Headline has no meaning! What is meant with “….targeting macrophage associated medicine”?
Response:
We had corrected as below: The possible therapy of osteoporosis by targeting macrophage. See page 9, line 316
Page 9, lane 319: What mans “… and TNF-a are regulated and the progression of bone loss is improved?
Response:
We had corrected as below: When macrophage-related cytokines, which include IL-1β, IL-6, and TNF-α, are regulated, the progression of bone loss is improved. See page 9, line 344-345
Page 9, lane 320: What means “When T- and B- cells are regulated, the activated macrophages are suppressed”??
Response:
We had corrected as below: When the function of B and T cells are regulated, the suppression of macrophage may develop through inhibition of IFN-γ. See page 9, line 345-346
Page 9, lanes 324-326: If Il-27 inhibits apoptosis of osteoblasts how can anti-IL-27 prevent loss of bone and preserve bone?
Response:
We had corrected as below: IL-27 from macrophages can suppress the expression of RANKL and inhibit the apoptosis of osteoblasts. Treatment with IL-27 agent prevented the loss of trabecular micro-architecture and preserved cortical bone parameters. See page 10, line 350-352.
Figure 1: osteocytes do not proliferate anymore they are terminally differentiated cells!
Response:
We had corrected the term of differentiation and showed as Figure 1.
Reviewer 2 Report
The authors have done a great job studying the role of macrophages in osteoporosis. There are however certain issues that need to be addressed before the article could be published.
- some phrases are hard to follow and therefore it is recommended that the whole text would be edited. Special attention is needed to long sentences and prepositions. There are also some spelling mistakes (RNAKL)
Line 252 verb is missing
Line 262 needs to be rewritten
Line 296 efficacy is not used correctly
- the article is about osteoporosis. I don’t know the reason the authors have mentioned the changes in RA patients and other diseases frequently in the text
- line 249, what do you mean with physiological function for bone resorption and formation
- what do you mean by the course of bone turnover
- conclusion section is missing
Author Response
The authors have done a great job studying the role of macrophages in osteoporosis. There are however certain issues that need to be addressed before the article could be published.
- some phrases are hard to follow and therefore it is recommended that the whole text would be edited. Special attention is needed to long sentences and prepositions. There are also some spelling mistakes (RNAKL)
Response:
Abbreviations: RANK, Receptor activator of NF-κB; RANKL, Receptor activator of NF-κB ligand; M-CSF, Macrophage-colony stimulating factor; TGF-β, Transforming growth factor-β; IFN-γ, Interferon-γ; TNF-α, Tumor necrosis factor-α; OPG, Osteoprotegerin; PTH, Parathyroid hormone; RA, Rheumatoid arthritis; TRAF, Tumor necrosis receptor-associated factor. See page 10, line 369-372
Line 252 verb is missing
Response:
We had corrected as below: Consequently, the RANKL/OPG ratio increases, osteoclastogenesis occurs, and the enhancement of bone resorption in the unloading activity is seen. See page 8, line 261-262.
Line 262 needs to be rewritten
Response:
We had corrected as below: Sclerostin from osteocytes may suppress bone formation by decreasing osteoblasts related autophagy or apoptosis in multiple myeloma-related osteolytic bone diseases. See page 8, line 285-287.
Line 296 efficacy is not used correctly
Response:
We had corrected as below: Bisphosphonates are most commonly used for the therapy of osteoporosis due to their effects in the suppression of osteoclast-mediated bone resorption. See page 9, line 321-322
- the article is about osteoporosis. I don’t know the reason the authors have mentioned the changes in RA patients and other diseases frequently in the text
Response:
RA is a systemic inflammatory disease with involvement of multiple joints. During the disease course related bony erosions, the elevation of circulating inflammatory cytokines including IL-1, IL-6 and TNF-α is observed. Higher prevalence of osteoporosis is usually found in RA patients when compared with general population. Different anti-inflammatory biologics are used to control the systemic inflammation. Therefore, we mention the effect of anti-inflammatory from these experiences.
- line 249, what do you mean with physiological function for bone resorption and formation
Response:
Osteocyte may promote bone resorption for bone loss, and may induce bone formation in different state.
- what do you mean by the course of bone turnover
Response:
Bone turnover refers to the total volume of bone that is both resorbed and formed over a period of time, usually expressed as percent/year. Bone turnover can be estimated by measuring relevant bone biomarkers.
- conclusion section is missing
Response:
We had added a conclusion section as below: Activated macrophages may secret proinflammatory cytokine that induce bone loss by osteoclastogenesis, and are associated with activation of bone resorption. Furthermore, adequate activation of macrophages is needed during the state of bone repair and is also associated with bone formation. Different expression of activated macrophages including proinflammatory mediators in M1 and anti-inflammatory reactions in M2, can regulate the progression of bone resorption and bone formation. Osteoporosis is always associated with systemic inflammation via the abnormal activation of osteoclasts. Adequate inhibition of macrophages may improve the systemic inflammation associated with osteoporosis. Macrophages can affect osteoclasts, osteoblasts, and osteocytes during the progression of bone loss. Regulation of proinflammatory cytokine from macrophages may inhibit osteoclastogenesis. Adequate stimulation of IL-27 from macrophages may inhibit osteocalstogenesis. IL-4 and IL-23 may induce macrophages to form multinucleated giant cells during the phase of bone resorption. The pharmaceutical targets over macrophage polarization or macrophage related cytokine dysregulation may be an important issue to develop in the future. See page9, line 355-368.
Round 2
Reviewer 1 Report
The authors have responded to my criticisms, which has considerably improved the manuscript in style and content. However, there are still several spelling errors and wrong phrasing throughout the manuscript.
Reviewer 2 Report
The authors have revised the articles accurately. However I still think the term "course of bone turnover" or "course of bone resorption" is not correct and the authors should simply use "during bone turnover/bone resorption"
And maybe instead of "physiological function for bone resorption and formation" you can use "function during bone resorption and formation